# The Role of Temperature on Inflammation and Coagulation: Should We Apply Temperature Treatments for Hemophilic Arthropathy?

**DOI:** 10.3390/ijms26052282

**Published:** 2025-03-04

**Authors:** Alice Maria Brancato, Laura Caliogna, Alessandra Monzio Compagnoni, Elena Cornella, Camilla Torriani, Micaela Berni, Liliana De Felice, Eugenio Jannelli, Mario Mosconi, Gianluigi Pasta

**Affiliations:** 1Orthopedics and Traumatology Clinic, IRCCS Policlinico San Matteo Foundation, 27100 Pavia, Italy; alicemariabrancato@gmail.com (A.M.B.); alessandramonzio.c@gmail.com (A.M.C.); elena.cornella01@universitadipavia.it (E.C.); eugenio.jannelli@unipv.it (E.J.); mario.mosconi@unipv.it (M.M.); gianluigipasta@yahoo.it (G.P.); 2Department of Clinical, Surgical, Diagnostic and Pediatric Sciences, University of Pavia, 27100 Pavia, Italy; micaela.berni@hotmail.com; 3Department of Public Health, Experimental and Forensic Medicine, University of Pavia, 27100 Pavia, Italy; camilla.torriani01@universitadipavia.it; 4Department of Medical-Surgical Pathophysiology and Transplants, University of Milano, 20122 Milano, Italy; liliana.defelice@policlinico.mi.it

**Keywords:** hemophilia, cold temperature, coagulation, heat temperature and hemophilic arthropathy

## Abstract

Hemophilic arthropathy (HA) is a complication of hemophilia, which is a genetic disorder characterized by a deficiency in blood clotting factors. HA is characterized by joint damage with inflammatory responses, pain, and movement limitations due to recurrent bleeding in the joints. The inflammatory reactions contribute to the activation of coagulation factors, which can exacerbate bleeding and further damage the affected joints. Therefore, the interaction between inflammation and coagulation plays a crucial role in the progression and complications of HA. Management strategies often focus both on inflammation and coagulation to alleviate symptoms and preserve joint function. Temperature can influence the inflammatory response and coagulation. The aim of this work was to understand how temperature management can positively or negatively influence the HA. We have carried out a narrative review of the available literature. This review explores the impacts of temperature on biological processes, and it discusses the possible clinical implications for the HA treatment. Our research shows that cold exposure has anti-inflammatory and analgesic effects, while heat is linked to pro-inflammatory cytokine release. Both hot and cold treatments are ill-advised for hemophilia patients. Heat stimulates neo-angiogenesis, and cold hampers coagulation, posing risks for increased bleeding in individuals with hemophilia.

## 1. Introduction

Hemophilia A and B are X-linked bleeding disorders caused by deficiency or absence of coagulation factors VIII and IX, respectively, with spontaneous bleeding. Repeated bleeding into the same joint (hemarthrosis) is responsible for hemophilic arthropathy (HA) characterized by joint disruption, pain, progressive rigidity, and limitation of motion [1].

Repeated bleedings in the joint lead to the development of synovial hyperplasia and angiogenesis, which further perpetuate inflammation and new vessel formation, sustaining this deleterious vicious cycle. The inflamed synovium is responsible for the production of lysosomal enzymes and pro-inflammatory cytokines. Both inflammatory and degenerative mechanisms (iron-dependent) contribute to articular cartilage degeneration in HA. It is well known that the presence of blood or hemo-derivatives in the joint can induce an inflammatory response, determining cartilage degeneration and dysfunction. Hemo-components are quickly cleared by resident macrophages, but when their clearance capabilities are overwhelmed because of recurrent bleedings, massive hemosiderin complex formation occurs [2]. The incorporated iron in synovial macrophages induces the secretion of pro-inflammatory cytokines such as IL-1β, IL-6, and TNF-α, which causes the release of nitric oxide, matrix metalloproteases and other catabolic factors [3]. The inflamed, hypoxic, and hypertrophic synovium stimulates the release of vascular endothelial growth factor (VEGF) with an abnormally greater vascularization [1]. Neo-angiogenesis supplies oxygen and allows the migration of inflammatory cells to the hypertrophic synovium, sustaining and perpetuating the inflammation [4]. In addition, it has been observed that the excess of intracellular iron has deleterious effects. For example, in cartilage it induces the production of damaging ROS, mitochondrial dysfunction and extracellular matrix degradation, promoting a metabolic shift of the affected chondrocytes and cartilage degradation [5,6,7].

On the other hand, it has been observed in hemophilic patients a reduction in the activation of the extrinsic arm of coagulation cascade, due to the high levels of tissue factor pathway inhibitor (that reduce the activation of the extrinsic coagulation pathway) and the low levels of tissue factor. Furthermore, in the synovial fluid of HA, a high amount of thrombomodulin, a cofactor involved in the activation of the anticoagulant protein C, has been found [1]. Thrombomodulin is anchored on endothelial cells and suppresses thrombus formation by preventing the binding between thrombin and its substrates, which are fibrinogen and fV. At the same time, thrombomodulin promotes the interaction between thrombin and protein C, causing its activation (Activated Protein C-APC). Then, APC can degrade fV and fVIII, contributing to the inhibition of thrombin generation and the coagulation process [8].

Currently, the treatment of hemophilic arthropathy is focused on the reduction in pain, functional recovery, and preventing further joint damage. The main and most effective treatment is factor VIII or IX replacement in association with all available strategies: anti-inflammatory drugs (not NSAIDs), hyaluronic acid injections, and analgesics [9]. To preserve the state and well-being of the joints, factor VIII or IX replacement therapy is performed as a prophylactic therapy.

Recently new treatments for the management of HA have been introduced, including the inhibition of fibrinolytic system with anti-plasmin antibodies or injection of small interfering RNA directed towards protease-activated receptors to avoid plasmin-induced cartilage damage. Some studies focus on anti-IL6R antibodies or anti-TNFα antibodies to reduce synovial hyperplasia and macrophage infiltration, while others concentrate on new potential strategies with MSC to reduce synovial thickness and vascularity.

The limitations of these proposed treatments are that they are only experimental, and none are used in clinics.

If synovitis does not disappear, surgical, chemical, or radiation-induced synovectomy is necessary to reduce the progression of joint damage. Radiosynovectomy is a minimally invasive, efficient, and secure method that uses a radioactive material every six months to treat chronic synovitis. It is very effective for treating bigger joints, such as knees, while chemical synovectomy is preferable for treating small joints, such as elbows and ankles. Chemical synovectomy is a weekly injection of rifampicin or oxytetracycline, and it is very widespread in developing countries due to low prices and the difficulty of recovering radioactive materials. In addition to chemical and radio synovectomy treatments, surgical intervention through arthroscopic synovectomy is also possible to preserve movement and joint function, as well as reduce pain and hemarthrosis [10].

## 2. Materials and Methods

### 2.1. Identifying the Research Question

The research question identified for the literature review was to investigate the role of temperature (both heat and cold) on inflammation and coagulation in hemophilic joint management, two important aspects to consider for possible effective treatments of HA are thermotherapy or cryotherapy.

### 2.2. Identifying Relevant Studies

A literature search was conducted to find relevant studies on the topic, which were identified through Pubmed and Scopus databases.

#### 2.2.1. Electronic Database Search

The following electronic databases were searched, taking into consideration the chronological span between 01 January 1993 and 08 August 2024:− PubMed− Scopus

The research strategy was carefully designed in order to retrieve the most relevant results. Due to the specificity of the two databases employed, a different search string was built for each one (1: PubMed search string; 2: Scopus search string). In brackets, the number of the results is given.

(Hemophili*[Title/Abstract] OR Haemophili*[Title/Abstract] OR Hemophilia[MeSH Terms] OR Haemophilia[MeSH Terms] OR Bleeding disorder*[Title/Abstract] OR Bleeding disorder[MeSH Terms]) AND (Joint diseases[Title/Abstract] OR Joint diseases[MeSH Terms] OR Arthropath*[Title/Abstract] OR Arthropathy[MeSH Terms] OR Synovitis[Title/Abstract] OR Synovitis[MeSH Terms] OR Inflammation[Title/Abstract] OR Inflammation[MeSH Terms] OR Coagulation[Title/Abstract] OR Coagulation[MeSH Terms]) AND (Temperature[Title/Abstract] OR Temperature[MeSH Terms] OR Cryotherapy[Title/Abstract] OR Cryotherapy[MeSH Terms] OR Thermotherapy[Title/Abstract] OR Thermotherapy[MeSH Terms] OR Heat Treatment[Title/Abstract] OR Cold Treatment[Title/Abstract]) AND English[la] AND (“1993/01/01”[Date-Publication]: “2024/08/08”[Date-Publication]) [243]TITLE-ABS-KEY(Hemophili* OR Haemophili* OR “Bleeding disorder*”) AND TITLE-ABS-KEY(“Joint diseases” OR Arthropath* OR Synovitis OR Inflammation OR Coagulation) AND TITLE-ABS-KEY(Temperature OR Cryotherapy OR Thermotherapy OR “Heat Treatment” OR “Cold Treatment”) [171]

#### 2.2.2. Other Sources

Eight studies were also included, according to the inclusion criteria. Article details are reported in Figure 1.

### 2.3. Study Inclusion Criteria

Starting from the research question, inclusion and exclusion criteria for the objective selection of the studies identified were defined as follows. Only studies published in the English language between 01 January 1993 and 08 August 2024 were eligible for inclusion.

### 2.4. Data Extraction

A standardized data extraction sheet was prepared, where the main information of the studies was collected (e.g., first author’s name, study title, publication year, and DOI). Titles, abstracts, and full texts were screened by the research team—i.e., two authors performed the study selection and the data extraction independently, and any disagreements were discussed between the authors.

### 2.5. Study Selection

Via the literature search, thirty-nine studies were included in this literature review, thirty-one of them were identified via database searches and eight via websites or citation searching (Figure 1). Figure 1 shows the process of study selection in detail, covering the number of search records retrieved from the two database searches (n = 414) and all other searches (n = 46), as well as the number of studies finally included (n = 39).

## 3. Results and Discussion

### 3.1. High Temperature Effects

Thermotherapy is the application of heat to a specific anatomical area resulting in increased tissue temperature.

Physiological heat therapy effects include pain relief, increases in blood flow, metabolism, and elasticity of connective tissue. Neural transduction of heat is mediated by TRP vanilloid 1 (TRPV1) receptors, which are ion channels activated by noxious stimuli. Activation of TRPV1 receptors within the brain may modulate anti-nociceptive descending pathways [11].

Some articles in the literature in the last decades study the role of hyperthermia on inflammatory processes. They show that whole-body hyperthermia induces high plasma levels of granulocyte-colony stimulating factor, IL-6, IL-8, IL-10, and IL-1β, thus promoting inflammation [12,13].

In particular, IL-6 is synthesized in injured sites in the initial stage of inflammation, and it contributes to the stimulation of acute phase response. While IL-6 acts as a defense mechanism by promoting the resolution of the injuries, it also enhances the transition from acute to chronic inflammation by changing the nature of leucocyte infiltrate from neutrophils to macrophages and promoting angiogenesis. For this reason, IL-6 has a dual effect, acting as both an anti-inflammatory and pro-inflammatory agent. Dysregulated continual synthesis of IL-6 plays a pathological role in chronic inflammation, so it would be interesting to investigate IL-6 blockade as an approach to treat chronic inflammatory diseases [14,15].

In addition, hyperthermia is involved in the primary mechanism of platelets to start and maintain hemostasis.

There are few literature studies, both in vivo and in vitro, on the effects of hyperthermia. The studies conducted show both experimental protocols and results that are very different from each other, providing no clear indications to suggest whether the temperature increase may have positive therapeutic effects on the coagulation process [16,17].

### 3.2. Low Temperature Effects

Cryotherapy is the application of cold to a specific body area, decreasing the temperature of the contact area and adjacent tissues.

Physiological cold therapy effects include blood flow and edema reduction, and slow delivery of inflammatory mediators, reducing inflammation of the affected area. Decreasing tissue temperature also reduces the metabolic demand of hypoxic tissues.

Cold therapy induces a local anesthetic effect, referred to as cold-induced neurapraxia, by decreasing the activation of tissue nociceptors.

In the literature, there are conflicting results about the outcome of in vivo studies.

In the rat model, it seems that local cooling reduces TNF-α production [18]; this result is confirmed by some human studies that show a decrease of pro-inflammatory cytokines such as IL-1β, IL-6, IL-17, and VEGF [19,20,21]. However, other studies conducted in humans do not confirm these experimental data because they do not find differences in whether cold is applied or not [22].

An interesting study shows that the combination between two different cold temperatures induces a significant increase in IL-1β and IL-4 (pro-inflammatory cytokine) and at the same time increases IL-2 and IL-10 (anti-inflammatory cytokine) [22]. The presence of both types of cytokines could have a positive effect, since pro-inflammatory cytokines mediate the acute inflammatory response that stimulates tissue repair and regeneration, while anti-inflammatory cytokines avoid the progression of the acute phase into a chronic phase. It would be interesting to further investigate the cytokine expression timing to understand the possible clinical endpoints [23,24].

Some authors have reported that local cryotherapy attenuates joint inflammation, pain, and articular swelling. However, other authors have stated that low temperature is associated with impairment of coagulation enzyme activity, hemostasis, and platelet function [25,26]. The literature also shows conflicting data [27,28].

Given that the literature contains different articles with conflicting data on the role of heat and cold in the inflammation and coagulation processes, we have summarized the articles reviewed in this work in the Table 1 below to try to outline the experiments carried out and their results.

### 3.3. Clinical Aspects in Heat and Cooling Treatments

HA occurs in patients with severe and moderate hemophilia A and B after repeated bleeding in a major joint, unless treated with replacement of the missing factor.

For a long time, physical treatments involving thermotherapy have been part of clinical practice for the management of arthropathy in hemophilic patients. Recently, we wanted to confirm their effectiveness through studies and laboratory research.

#### 3.3.1. Cold Treatment

The protocol “R.I.C.E.” (Rest, Ice, Compression, Elevation) and cryotherapy are commonly used for initial treatment of acute musculoskeletal injuries and bleeding.

These protocols are also used to treat hemorrhagic arthropathy in hemophiliac patients because they carry a minimal risk.

Cryotherapy is currently the most recommended method to relieve acute pain in hemarthrosis. The mechanism for pain relief is thought to be a combination of reduced local nerve conduction rate and central mechanisms mediated in the spinal cord.

In addition, local cold application can stimulate the skin’s cold receptors, which can then act on the pain gate to reduce pain perception. Based on a literature review, it emerged that crushed ice is probably the safest method of application, the application should not exceed 20 min at two-hour intervals and should always be guided by pain and discomfort levels. The literature has shown that there are also other modes besides ice, including gel packs and cryo cuffs.

Cryotherapy also seems to be effective for the treatment of tissue lesions having a role on the cellular activity, in fact it inhibits the metabolic activity of the damaged tissues.

This reduction preserves the cells from secondary cellular damage and possibly from apoptosis due to ischemic environment conditions that arise in the post-lesion phases.

However, recent studies do not confirm that ice improves the overall outcome or that it stops hemorrhage or blood vessel swelling. Studies in people with hemophilia (PWH) show that the application of ice seems to increase the morbidity of bleeding, further compromising coagulation and hemostasis.

Local application with ice, both before and immediately after medical management, in PWH during the acute blood vessel would appear to impair hemostasis and coagulation in the damaged synovial membrane. This impairment can cause prolonged bleeding, which in PWH may lead to potentially increased amounts of intra-articular blood accumulation during an acute arthropathy.

In support of this thesis, Forsyth et al. published a study in 2012 that demonstrated how experimental cooling of blood and/or tissues, both in vitro and in vivo in human and animal models, compromises coagulation and prolongs bleeding. Forsyth reported that application of ice may reduce pain during the treatment of acute blood pressure in vivo, but it is associated with prolonged bleeding leading to increased blood in the joint, resulting in blood-induced joint disease. In other words, LC (local cryotherapy) had the potential to increase the morbidity of bleeding by further compromising coagulation and hemostasis [29].

In addition, Forsyth et al. published another article in 2012, in which they questioned the effectiveness of LC as a treatment for hemorrhagic arthropathy in hemophiliac patients (AHH), because the low temperature was associated with impaired enzyme activity of coagulation and platelet function. Ice potentially leads to an increase in blood in the joint with its associated deleterious consequences, especially in those situations where an infusion of fVIII or fIX, used as a hemophilic treatment, is delayed or unavailable [30].

Although ice can help manage acute pain related to hemarthrosis, other interventions are available that will not affect coagulation and hemostasis.

Cryotherapy and local application of ice seem to prolong the bleeding time; nevertheless, the application of ice continues to be a universal recommendation in people with hemophilia and remains an additional treatment in those realities where access to coagulation factor concentrate (CFC) or other substitute therapies is possible.

This idea is supported by some studies in the literature, including the study that Seuser et al. published in 2007, which recommends the use of LC for the rehabilitation of synovial patients with AHH [31].

Another study supporting cryotherapy is an article by Ribbans et al. which suggests a combination of rest, coagulation factor replacement, LC, physical medicine, and supervised rehabilitation as an ideal treatment for AHH [32].

An interesting study by Young et al. published in 2008, recommends the use of cryo cuff devices to generate LC for the treatment of hemorrhagic arthropathy in hemophilic patients. All patients reported that the device had a significant impact on pain relief and return to pre-blood condition. In addition, 78% of patients felt that the cryo cuff led to a significant reduction in inflammation around the joint. There was no conclusive evidence that the cryo cuff caused a decrease in the amount of factor used to treat AHH. No other adverse clinical effects were found in this study [33].

Two other studies, one conducted by Ravanbod et al. and the other by Gallo et al., reported that LC could diminish swelling and temperature in AHH, and Ultrasonography demonstrated a positive effect of LC on joint inflammation [34,35].

#### 3.3.2. Heat Treatment

On the other hand, physical heat treatments are not as widespread and standardized in the treatment of hemophilic hemorrhagic arthropathy. Generally, they are not used in the acute phases but during recovery alongside physiotherapy.

Currently, the most widely used heat therapy for the treatment of HA is High intensity pulsed laser therapy (HILT).

Preliminary studies conducted in 2018 by El-Shamy et al. and Demartis et al. report that HILT is a potentially effective recent therapy for pain relief, postural control enhancement, and weight-bearing pattern adjustment in children with hemophilia [36,37].

These results were confirmed more recently by Elnaggar et al., who, through a study conducted on 40 children, showed a positive effect of laser therapy in reducing pain in children with hemophilic arthropathy [38].

Both studies recommended the use of a High intensity pulsed Nd:YAG laser instead of other types of lasers because HILT allows for penetration and easier spread within the tissue, reaching deep into the joint to be treated and covering a large area in a short time. Furthermore, the studies recommended laser therapy in association with physical exercises.

Some interesting studies in the literature have tried to identify a correlation between high temperature and inflammation.

In particular, Etulain et al. observed that exposure to high temperatures seemed to prevent IkB degradation with the consequent inhibition of the transcription factor NF-kB.

NF-kB is involved in the activation of genes encoding pro-inflammatory cytokines (IL-1β, IL-6, IFN-γ, TNF-α) [16]. Studies conducted on murine models of HA showed that the levels of NF-kB and its inflammatory mediators progressively increased in damaged joints and increased synovitis damage [39].

Therefore, it seemed that heat indirectly inhibits the NF-kB pathway by relieving inflammation, suggesting its potential use in hemophilic patients. However, more detailed studies are needed to verify this hypothesis.

Other studies have instead observed that heat treatment seemed related to a decrease in coagulation and an increase in inflammation [12,13,17], suggesting that high temperatures could worsen HA symptoms.

Currently, there are only a few in vivo studies on hemophilic patients regarding the use of thermotherapy, which suggest that high temperatures reduce joint pain in patients. It would be interesting to deepen and increase these studies to understand whether high temperatures, in addition to inhibiting pain, cause a worsening of joint inflammatory state and have a negative effect on hemophilic patients’ coagulation.

In this review, we tried to examine the role of hot and cold temperatures both on coagulation and inflammation. Unfortunately, few articles and reviews in the literature deal with this topic, especially related to hemophilic patients. Luckily, the actual information about this topic mostly derived from human in vivo studies.

Our results demonstrated that the heat treatment may have deleterious effects both at inflammatory and hemostatic level: indeed, it came out that thermotherapy induced a significant release of pro-inflammatory cytokines such as IL-6, IL-8, IL-10, and IL-1β. Furthermore, it increased vascularization, causing endothelium damage and inducing VEGF expression [12,13,16]. However, the growth of pro-inflammatory cytokines and vascularization was deleterious for the management of hemophiliac arthropathy because it increased bleeding and synovial damage.

On the contrary, it has been shown that cold temperature promotes the production of anti-inflammatory cytokines in rats [18,19], even if studies in humans showed either no difference in inflammation after cold water immersion [20] or the release of both pro and anti-inflammatory cytokines [22]. If it is true that cold attenuates chronic inflammation, ice could be used as a treatment for hemophilia. However, it has been observed that cold prolongs coagulation time and promotes bleeding, representing a potential risk for hemophilic patients; therefore, different studies are recommended to gain more information about the cold–coagulation relationship.

## 4. Conclusions

In conclusion, hot temperature seems to have worse consequences than cold temperature, because cold has anti-inflammatory and analgesic effects, while heat is associated with the release of pro-inflammatory cytokines such as IL-6, IL-8, IL-10, and IL-1β. Since in hemophilia there is an increase of these cytokines, hemophilic patients should avoid hot temperatures to avoid sustaining inflammation.

Furthermore, heat can stimulate neo-angiogenesis and new vessels allow the migration of inflammatory cells to the hypertrophic synovium, perpetuating the inflammation. However, cold impairs coagulation and can promote bleeding, which is problematic for hemophilic patients, whose coagulation is already compromised. Then, based on these studies in the literature, both hot and cold exposure (Figure 2) should be inadvisable for the treatment of hemophilia patients. However, other studies are necessary to verify the precise role of temperature to better understand how it could be used for the treatment of hemophilia.

## Figures and Tables

**Figure 1 ijms-26-02282-f001:**
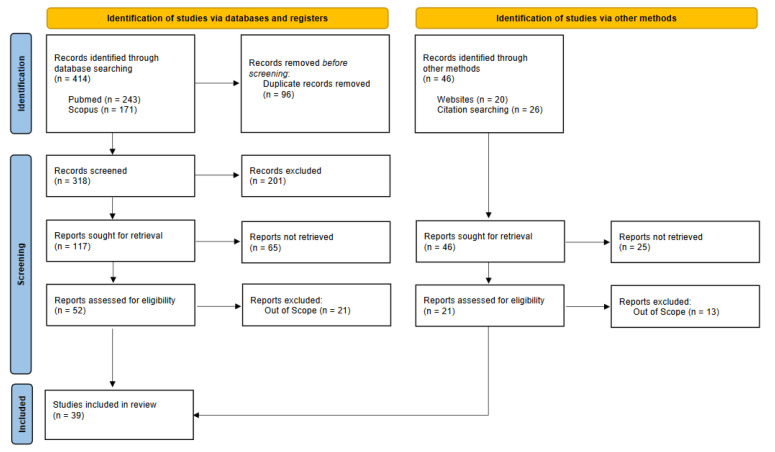
Prisma 2020 flow diagram for new systematic reviews which included searches of databases, registers, and other sources.

**Figure 2 ijms-26-02282-f002:**
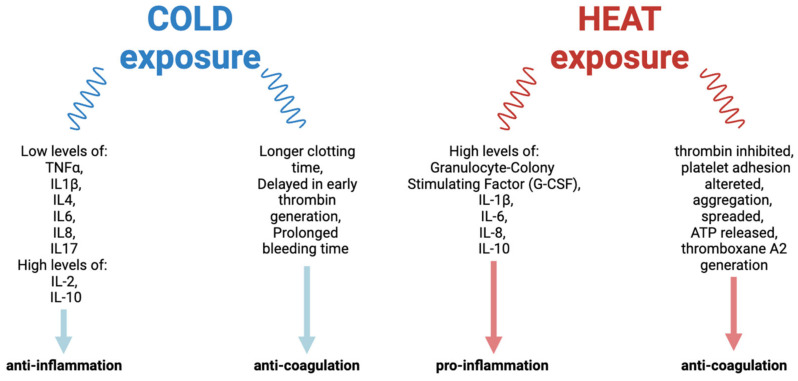
Summary of the main effects of cold exposure and heat exposure.

**Table 1 ijms-26-02282-t001:** Summary of the studies analyzed.

Study	Type	Exposure Temperature	Time of Treatment	Time of Measurement	Specific Parameter(s) Measured	Outcome
Heat Exposure						
[17]	human	48–52 °C	90′	before, 0.5 h and 2 h post	angiogenic and inflammatory mediators and genes	promotes angiogenic mediators and increased capillarization
[13]	human	42 °C	20′	before, 0 h, 1 h, 2 h post	IL-6, TNF-α, hsCRP, CBC, total blood cell	increases leukocyte cell
[12]	human	41.8 °C	60′	before, during, 2.5 h, 5.5 h, 23.5 h post	cytokines	increases pro-inflammatory cytokines
[16]	human	38.5–42 °C	10′	-	platelet behavior	negative effect on platelet hemostatic function
Cold Exposure						
[28]	human	33 °C	-	before, 22 h, 48 h post	thromboelastometry	difference in coagulation
[25]	human	33 °C	-	24 h, 48 h post	blood cell count, INR, APTT, thrombin time, fibrinogen, blood lactate employing, CRP	negative effect on thrombin generation
[26]	human	27–32 °C	20′	20 min post	thrombin generation, fibrin clot analysis, TEG, fibrinogen consumption, factor XIII activation	problems in thrombin generation and clot formation
[22]	human	27 °C, 7 °C, 27 °C	40′, 30′, 2 h	40 min, 30 min, 2 h post	heart rate, VO_2_, EE, shiver, inflammatory cytokines, acute phase proteins, lipid biomarkers	induces proinflammatory cytokines, increased anti-inflammatory cytokines
[27]	human	18 °C	within 20′	-	TEG analysis	doubled bleeding time
[20]	human	10 °C	10′	before, 2 h, 24 h and 48 h post	creatine kinase, plasma cytokines, GDNF, NGF, HSP70 expression	no influence in inflammation pathway
[19]	animal	4 °C	1–7 days	1–7 days	catecholamines production, inflammatory gene expressions relate to adipocyte browning	induces UCP-1 expression and anti-inflammatory response
[21]	human	Ice 30 °C intra-joint	30′	before, 24 h later	IL-6, IL-1β, TNF-α, IL-17A, VEGF, NF-kB-p65 protein, and PG-E2	reduces IL-6, IL-1β, and VEGF synovial protein levels
[29]	human	crushed ice, ice bag	15–60′	-	-	impairs hemostasis and coagulation
[18]	animal	Δt = ±10 °C	180′	-	cytokine and blood flow patterns	reduces soft tissue damage

## Data Availability

The data that supports the findings of this study are available within the article.

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
