# Peer review of "The Role of Temperature on Inflammation and Coagulation: Should We Apply Temperature Treatments for Hemophilic Arthropathy?"

_ijms, 2025, doi:10.3390/ijms26052282_

Round 1
Reviewer 1 Report
Comments and Suggestions for Authors
The authors have a worthy goal which is to address recommendations as to the use of heat or cold in joint bleeds in patients with hemophilia. The challenge with this manuscript is that the vast majority of the manuscript focuses on the general concepts of heat and cold effect in a variety of situations that may or may not be particularly applicable to hemarthrosis. For example, the discussion of heat stroke as a model for understanding the potential treatment of hemarthrosis seems inappropriate. The same could be said for hypothermia after cardiac arrest.
1. The authors should separate out the general studies of the biologic effects of heat or cold from the small number of actual studies addressing the use of heat and cold in hemophilia patients. Perhaps a better way to structure this review would be to substantially compress the discussion of the studies that are outlined in Table 1 that are not specifically relevant to hemarthrosis; then expand the discussion of the studies specific to hemophilia such as references 34, 35, 37, 39. The authors can then move to their conclusion that there is not sufficient data to support the use of either of these modalities in hemophilia joint bleeds. This would be a much shorter manuscript.
2. The most significant element of the review as it currently stands is Table 1 although Figure 2 is also a nice useful summary of general biology. Table 1 requires substantial revision. Going column by column, the studies should be cited by reference number or at least the reference number should be included since a number of the named authors for the studies occur in more than one reference. In terms of study type, all are listed as "in vivo". It would be more useful to list them as human or animal. An additional column should be added indicating the specific parameter(s) measured. The final column "Outcome", needs to be modified to say exactly what is meant by "pro-angiogenesis" for example. One way to do this could be to have something like "Pro-angiogenesis: ..." then state the specific parameter change.
Author Response
We are grateful for your valuable comets and advices.
As suggested by the reviewers we have made the following changes:
- The authors should separate out the general studies of the biologic effects of heat or cold from the small number of actual studies addressing the use of heat and cold in hemophilia patients. Perhaps a better way to structure this review would be to substantially compress the discussion of the studies that are outlined in Table 1 that are not specifically relevant to hemarthrosis; then expand the discussion of the studies specific to hemophilia such as references 34, 35, 37, 39. The authors can then move to their conclusion that there is not sufficient data to support the use of either of these modalities in hemophilia joint bleeds. This would be a much shorter manuscript.
Thanks for the advice, we rewrote the discussions following your suggestion and separating the general studies on heat or cold effects from the studies regard hemophilia patients
- The most significant element of the review as it currently stands is Table 1 although Figure 2 is also a nice useful summary of general biology. Table 1 requires substantial revision. Going column by column, the studies should be cited by reference number or at least the reference number should be included since a number of the named authors for the studies occur in more than one reference. In terms of study type, all are listed as "in vivo". It would be more useful to list them as human or animal. An additional column should be added indicating the specific parameter(s) measured. The final column "Outcome", needs to be modified to say exactly what is meant by "pro-angiogenesis" for example. One way to do this could be to have something like "Pro-angiogenesis: ..." then state the specific parameter change.
Thanks for the advice, we have corrected table 1 as the reviewer suggests.
Reviewer 2 Report
Comments and Suggestions for Authors
The study aims to highlight the role of temperature in the management of hemophilic arthropathy through depicting the impact of heat and cold exposure on both inflammation and coagulation which are the main players in HA. overall, the review is clear, well-structured, and relevant to the field. The references are recently updated. The figures and table are appropriate and properly show the data. The conclusion is coherent and supported by listed citations.
Minor comments:
1- In the introduction, I suggest to add paragraph about the current management of HA and show the limitations of this protocol which promote looking for alternative methods (the gap). I also recommend adding a paragraph of using Temperature in current disease management (if applicable)
2- Please, list the meaning of any abbreviation before putting it in the text for fist place:
- AHH.. Page 7 line 287 , it has been listed in page 8.
- PWH..page 7 line 274, hasn't been mentioned at all.
3- Table 1, I suggest to specify the Spp. of study as one of them in rat.
4- Omit which in page 4 line 131.
Author Response
We are grateful for your valuable comets and advices.
As suggested by the reviewers we have made the following changes:
1- In the introduction, I suggest to add paragraph about the current management of HA and show the limitations of this protocol which promote looking for alternative methods (the gap). I also recommend adding a paragraph of using Temperature in current disease management (if applicable)
Thanks for the advice, we have add the paragraph about the current management of HA how the reviewer suggests.
2- Please, list the meaning of any abbreviation before putting it in the text for fist place:
- AHH.. Page 7 line 287 , it has been listed in page 8.
- PWH..page 7 line 274, hasn't been mentioned at all.
We correct it
3- Table 1, I suggest to specify the Spp. of study as one of them in rat.
Thak for the advice, we correct the table 1.
4- Omit which in page 4 line 131.
We do it
Reviewer 3 Report
Comments and Suggestions for Authors
Dear editors,
Thank you for giving me the opportunity to review this important research entitled “The Role of Temperature on Inflammation and Coagulation: A Possible Hemophilic Arthropathy Treatment. A Systematic Review”. It is a great honor and pleasure for me to be invited as the reviewer for this topic. Alice Maria Brancato and the co-authors have comprehensively reviewed the temperature effects on Inflammation and Coagulation in Hemophilic Arthropathy. This topic of study holds significant importance, as it is attributed to their team’s long-term efforts and contributions in the scientific field. Although the article is well-written, I have a number of comments concerning this study:
1. Title: In light of the conclusion that both hot and cold treatments are ill-advised for hemophilia patients, the sentence” A Possible Hemophilic Arthropathy Treatment” could be deleted and replaced by “Shall we apply temperature treatments for Hemophilic Arthropathy?” to arouse readers' interest.
2. Line 300: Following a systematic review of the literature, crushed ice is probably the safest method of application, but other modalities including gel packs and cyrocuffs can be used. Application should not exceed 20 minutes at two hourly intervals and always be guided by levels of pain and discomfort. I wonder if there is objective guidance or optimal time course limitation?
The research is interesting and novel, refreshing our understanding of temperature effect on HA treatment in this vulnerable population. From the perspective of health care, the article should be published as soon as possible after minor revision.
Author Response
We are grateful for your valuable comets and advices.
As suggested by the reviewers we have made the following changes:
- Title: In light of the conclusion that both hot and cold treatments are ill-advised for hemophilia patients, the sentence” A Possible Hemophilic Arthropathy Treatment” could be deleted and replaced by “Shall we apply temperature treatments for Hemophilic Arthropathy?” to arouse readers' interest.
We correct it
- Line 300: Following a systematic review of the literature, crushed ice is probably the safest method of application, but other modalities including gel packs and cyrocuffs can be used. Application should not exceed 20 minutes at two hourly intervals and always be guided by levels of pain and discomfort. I wonder if there is objective guidance or optimal time course limitation?
Currently there are very few articles in the literature on the use of cryotherapy as a treatment for joint pain, and some have conflicting data, so the available data are a recommendation and there is no objective guide yet.
Round 2
Reviewer 1 Report
Comments and Suggestions for Authors
The authors have been responsive to reviewer comments, resulting in improved manuscript.